# Prioritization of the Best Sustainable Supply Chain Risk Management Practices Using a Structural Analysis Based-Approach

**Manel Elmsalmi [1], Wafik Hachicha [2],\* and Awad M. Aljuaid [2]**

[1] OLID Research Unit, Higher Institute of Industrial Management Sfax, University of Sfax, Technopolis of Sfax 3021, Tunisia; manel.elmsalmi@isgis.usf.tn

[2] Department of Industrial Engineering, College of Engineering, Taif University, P.O. Box 11099, Taif 21944, Saudi Arabia; amjuaid@tu.edu.sa

\* Correspondence: w.hashisha@tu.edu.sa or wafik.hachicha@isgis.usf.tn; Tel.: +966-53-194-0695

**Abstract:** Companies attempt to improve the performance of their supply chain (SC) by distinguishing and presenting feasible sustainable development practices (SDP). Considering SDP without focusing on sustainability risks may disturb the company's future. Very few studies in the extant literature have dealt with the impact of (SDP) on the supply chain risk management (SCRM). In fact, the aim of this paper is to classify and prioritize SDPs according to their priority for better risk management and effective SC performance. The proposed approach comprises two phases. First, 14 SDPs are identified and selected from the literature. Second, MICMAC (Matrice d'impacts croisés multiplication appliquée à un classement) method as a structural analysis method applies to identify and assess sustainable supply chain risk management (SSCRM) practices which reduce risk in the SC. The input data for each phase are based on Delphi technique, which is a process group used to collect the opinions of experts in the field. The aim of the proposed approach is to prioritize SSCRM practices and classify them into influential, non-influential, independent and dependent practices and their mutual relationships. The six key findings SSCRM practices from direct and indirect classification include the following elements: (1) Delayed differentiation, (2) Information sharing with upstream and/or downstream partners, (3) Simplification of product dismantling/anticipation of product end of life, (4) Supplier/subcontractor's performance assessment, (5) establishing shared supply management and (6) establishment of contracts with transporters.

**Keywords:** supply chain risk management; sustainable development practices; sustainable supply chain management; practice prioritization; sustainable supply chain risk management; strategic prospective; structural analysis; MICMAC

## 1. Introduction

Many factors largely determine the future, including the forces of nature, social, economic and political dynamics, scientific discovery and technological innovation. However, human choice increasingly determines the future. As a result, the society cannot fully control the future, but it can influence the course of history. So it is necessary to think about the balance between what we want and what is possible. Therefore, companies should track a strategic prospective to keep the sustainability of companies and their activities.

Then, the supply chain management (SCM) concentrates on integrating a firm's internal management processes with the external environment. This could explain why sustainability has been embraced by scholars who studied the SCM [1]. Many investigations featured on both sustainability and SCM in the common area named Sustainable Supply Chain Management (SSCM) [1–7]. It is imperative to give a hypothetical comprehension of the key exercises or practices associated with this idea in a manner to make the SSCM operational. Different authors studied and highlighted the SSCM practices to

include sustainable procurement, supplier partnerships, information sharing, sustainable distribution, sustainable packaging, reverse logistics, etc. for various purposes. For instance, Lis et al. [5] concluded that the main thematic areas in the SSCM research field included the following domains: (1) the economy and management in the environment's context, (2) the supply chain in sustainability, (3) the sustainable supply chain process approach, (4) the decision making for SSCM, (5) the practice context of the SCM and (6) the competition and social responsibility issues. According to Beske et al. [6], SSCM practices comprise strategic orientation, supply chain continuity and collaboration, risk management and proactivity for sustainability. Esfahbodi et al. [7], also highlighted the SSCM practices from sustainable production, sustainable design, sustainable distribution and investment recuperation viewpoints. From the above, it is apparent that the SSCM rehearses embraced by the analysts are affected by the reason for their investigations. This has added to the developing nature of SSCM practices. Baah and Jin [8] have described SSCM from the viewpoint of three core components of the SCM: the flow of materials, supplier partnership and coordination and information sharing.

This study incorporates four-dimensional sustainability by viewing the best key sustainable risk management practices. These components are adapted in this study because they largely capture the essence of the SSCM [9]. According to Baah and Jin [8], it is good to examine the individual SSCM practices on their performance, but it is better to examine the total effect by considering the composite of the SSCM practices on organizational performance. In addition, the supply chain risk management (SCRM) research has mainly mistreated the importance of sustainability issues [10–12]. There is little knowledge about risk sustainable management and how these practices influence firms' performance.

According to Giannakis and Papadopoulos [10], the primary goal of the SCRM is not only cost-saving, but also it can enhance the SSCM by creating values. Risks, such as natural disasters, energy consumption, packaging waste and environmental damage caused by logistics and transportation are considered as SR sustainability risks.

Recently, Hsu et al. [13] have developed an integrated quality function deployment-based approach which ultimately provides an example of a company with a useful approach for the development of resilient, sustainable supply chains. A few recent research studies have used various multi-criteria approaches to evaluate the sustainable supply chain risk management (SSCRM) practices [12,14–16]. Indeed, according to Syed et al. [17], sustainable internal business process risks, sustainable supply risks and sustainable demand risks have a negative relationship with the supply chain integration.

Therefore, the aim of this study is to prioritize the best selected sustainable risk management practices from the extant literature for an efficient SC performance. For this end, using the MICMAC method as a structural analysis tool, the inter-relationship between practices and their classification is analyzed, which might serve as avoiding supply chain risk mitigation strategies.

### 1.1. Background and Literature Review
### 1.1.1. The Sustainable Development (SD)

The notion of Sustainable Development (SD) had a long history in the last 50 years. Recently, the SD is the subject of a great deal of research. According to Linton et al. [18], the SD concerns very diverse disciplines, such as earth and environmental sciences, medicine, material sciences, agriculture, biology, economics, social sciences and management sciences.

According to the definition of the United Nations Conference on Environment and Development (UNCED), there are two main concepts inherent in the SD. First, development means that current activities must meet the needs of the present and can meet those of the future [19]. However, these activities are conditioned by the sustainability, which is a constraint as a way of ensuring that development can take place in the long term, but not just over a short period. This is the reason the subject of SD has received a lot of attention in all fields in recent years. Making the notion of SD operational in public policies involves several important challenges linked to the measurement of this concept. Indeed, in the

absence of indicators or a quantitative framework, policies in favor of the SD will lack a solid foundation on which they can build to move forward. In fact, the term SD can be analyzed according to three dimensions: economic, environmental and social united by a set of synergies and complex trade-offs.

### 1.1.2. The Sustainable Supply Chain (SSC)

With the emergence of the SD concept, logistics is no longer concerned only with the achievement of economic objectives but must also integrate environmental objectives. Thus, the concept of the sustainable supply chain (SSC) concerns both companies and political power.

According to Pagell and Wu [20], the SSC must be efficient on the three dimensions of performance; "The supply chain must perform compared to traditional measures of profit and loss, but also compared to broader conceptions of performance, including social and environmental dimensions." The authors also considered that "To be truly sustainable, a supply chain at worst would do no net harm to physical or social systems while still producing a profit over a long period; a truly sustainable supply chain must continue to do business forever".

The sustainable supply chain management (SSCM) can be seen as integrating environmental and social concerns into the management of the supply chain, by improving and balancing the three types of performance: economic, environmental and social. Like sustainable development, sustainable logistics is a discipline that considers economic, ecological and social constraints when making logistics decisions.

In this vein, Seuring et al. [21] defined "SSCM in their editorial as," the management of the flow of information and cooperation integrating economic, environmental and social objectives and the expectations of the stakeholders". Seuring and Müller [22] added physical and financial flows to the definition; "sustainable supply chain management is like the classic SCM, which includes the management of the three traditional logistics flows (physical flow, information flow and financial flow) as well as cooperation along the supply chain. However, to be sustainable, this SCM must also consider objectives in relation to the three dimensions of sustainable development (economic, environmental and social), arising from the expectations of customers and stakeholders [23]. Thus, supply chain partners need to be fully monitored and evaluated with key components of the SSCM which are: environmental, social and economic criteria.

In fact, the SSCM is defined as a well-structured, transparent combination and attainment of an organization's social, environmental and economic objectives in the planned coordination of key cross-organizational business processes, which facilitate the long-term economic outcomes of that peculiar company and its supply chains [24,25]. According to Peters [26], the most accepted definition for the SSCM is that "managing the SCM activities in relation to the environmental, economic and social issues to attain the long-term economic goals of individual organization and its supply chains". The SSCM activities that have been discussed in the literature review by Al-Odeh and Smallwood [27] are design, production, marketing, transportation and purchasing. A brief review has been provided for each activity as it is related to sustainability concept.

### 1.1.3. The Supply Chain Risk Management

The SCRM is defined as "the systematic identification of risks and implementing all measures necessary to limit exposure to risks. This term covers activities such as the data and information collection, the risk analysis and assessment, the measure prescription and execution, as well as the regular monitoring and evaluation of a process and its results, based on international, community and national sources and strategies".

In this context, Zsidisin and Ritchie [28] pointed out that the "SCRM is no longer a purely reactive activity of improving the organization's capacity to absorb disruption but it is also about preventive and collective activity seeking to preserve the creation of value in

potential circumstances". These two researchers specified that "the natural mission of the SCM is to create value through the chain so that the SCRM can preserve this creation".

Therefore, it can be drawn from these definitions is that the SCRM is not only about implementing all the measures limiting risks exposure but also about preserving the value which they have created and therefore the sustainability of the supply chain. This proves that the combination of these two themes prompts researchers to think about studying the SDP applied to SCRM.

Hofmann et al. [29] developed a transdisciplinary approach and provided a concise description of how sustainability issues in supply chains materialize as risks for focal firms. Building on this mechanism and drawing on the stakeholders' theory, a conceptualization of sustainable risks is developed which lays the basis for future investigations in this respective field. In addition, a viable management concept for sustainable supply chain risks is devised. The results acquired by Harclerode et al. [30] promote the consideration of sustainability and risk management concepts in all stages of the sanitation project life cycle to achieve a sustainable outcome. Wenyan [31] used a decision-making trial and the evaluation laboratory (DEMATEL) as an effective tool to identify critical issues in the SSCM and the interrelationships between the different risk factors.

Giannakis and Papadopoulos [10] developed a risk management approach based on the supply chain sustainability to explore the nature of sustainability-related supply chain risks and distinguish them from the typical supply chain risks and develop an analytical process using the Pareto method for their management. As for Valinejad and Rahmani [32], they proposed a comprehensive and credible framework for managing the sustainability risks of the supply chain for telecommunication companies, based on a novel approach to sustainability.

Abdul-Moktadir et al. [33] presented an analysis of risk factors in SSCM in an emerging economy of leather industry. As for Benabdallah et al. [34], they have presented an incorporated rough DEMATEL technique which considers the interrelationship between the various risks and the gathering inclination variety.

In fact, another way to study the SCRM and to identify the most critical risk is the use of structuring modeling tools, such as the Interpretive Structural Modelling (ISM) and the MICMAC methods. For instance, Jha and Devaya [35] modeled the risks faced by Indian construction companies assessing international projects. Liu et al. [36] identified risk factors of building apartments for University talent through the agent construction mode in China. In this vein, Hachicha and Elmsalmi [37] proposed an integrated approach based-structural modeling for risk prioritization in the supply network management. In this context, Jiang et al. [38] identified a significant risk and analyzing risk relationship for construction projects in China. As for Troche-Escobar et al. [39], they studied the SCRM in the Brazilian Wind Power Projects. Recently, Edgar et al. [40] had proposed an investigation about the Andean region of Peru for managing risk in the agri-food supply chain while Chen, et al. [41] identified decisive Socio-Political sustainability barriers in the supply chain of the banking sector in India.

### 1.2. Objective of the Study

The literature review analysis shows that the majority of previous studies in the extant literature have not provided direct evidence of the relationship between SCRM, SDP and SC performance at the same time. Therefore, to overcome the shortcomings of previous studies outlined above, this paper proposes to classify the best sustainable management practices that have the most important impact on the SCRM and for the entire supply chain performance. These can therefore avoid supply chain risk mitigation strategies for higher supply chain performance.

In fact, this research is the first step towards a more profound understanding of the SS-CRM practices through four criteria, such as (1) their direct influences classification, (2) their indirect influences classification, (3) their direct dependencies classification and (4) indirect dependencies classification. The idea behind this classification and this prioritization is

that practicing the most important sustainable risk management practices in the SSCM will improve the performance of the entire supply chains and will positively influence (directly and indirectly) the rest of the practices. The challenges with structural analysis using the MICMAC method are behind the growing relevance of determining the key system variables, especially in the SCRM. Therefore, the most important retained practices will require special attention regarding the development of risk mitigation strategies.

## 2. Materials and Methods

### 2.1. Strategic Prospective and Structural Analysis Using the MICMAC Method

2.1.1. Strategic Prospective

The prospective is a reflection on the future, which applies itself to describing the most general structures and to identify the elements of a method applicable to our speeding up world. In this vein, Godet [42] adds that "the prospective takes the form of collective reflection more and more often, a mobilization of minds in the face of the changes in the strategic environment". In fact, the purpose of prospective is to explore, create and test visions of both futures. Future visions can help produce long-term policies, strategies and plans. Bradfield et al. [43] consider it as "a formalized approach that uses a combination of qualitative and quantitative tools. The researchers describe it as a mixture of intuitive logic and PMT methodologies, changed probabilistic trends".

2.1.2. The Structural Analysis with the MICMAC Method

Therefore, to analyze a system, it is not enough to identify its components. In fact, the relationships between the system variables should be understood. In the literature there are two main approaches to analyze the systems: analytical approaches and systemic approaches. Contrary to analytical approaches, the systemic approaches principally apply for more complex systems. In this sense, Godet and Durance [44] show that in a systemic approach, a variable exists through its relationships. As for Godet [42], he specified that "Structural analysis consists in relating the variables in a double entry table (structural analysis matrix)". The filling of the structural analysis matrix is both qualitative and quantitative. Qualitative filling is used to identify all the direct influence relationships between the variables. For example, in the presence of variables B and C, it is advisable to ask the two following questions:

(1)    Does variable B have a direct influence on variable C?
(2)    Does variable C have a direct influence on variable B?

In fact, when there is no direct influencing relationship between two variables, it should be assigned a value of 0 to the structural analysis matrix while there is a relationship of direct influence between two variables, the relationship should be quantified by estimating its importance. With a strong influence relationship, the assigned value should be 3. Value 2 is assigned to a medium influence relationship while value 1 is assigned when there is a weak influencing relationship.

The structural analysis is inspired by graph theory and operations research simulation work carried out shortly after World War II in the United States, notably at the Rand Corporation for the needs of the United States military. It gives as exhaustive a representation as possible of the system and makes it possible to reduce its complexity to the essential variables, which amounts on the one hand to highlighting the key variables of the system, whether they are hidden.

Godet [42] defines structural analysis as "a method of structuring a collective reflection, the chosen project of which can be considered as a system and can be defined as a set of interacting elements". Godet and Durance [44] add that it is "a systemic method, in matrix form, for analyzing the relationships between the variables making up the system studied and those of its explanatory environment".

In fact, there are two ways of using structural analysis. The first one is about the decision-making use: research and identification of the variables and actors on which we

must act to achieve the objectives we have set. The second way concerns the prospective use: which is the search for the key variables on which the prospective reflection should focus.

Hatem et al. [45] stated that "a structural analysis is a system analysis technique which focuses on the studied field by proceeding with three successive questions". The first question is about how to identify the components of the system and the relationships between its components. The second question is about how the system works, while the third question outlines the evolution of the system.

The stages making up the structural analysis include the following items: (1) the identification of variable that are it is crucial for the rest of the process. Then, step (2), which is the least formal, identifies the relationships between the variables, and step (3) which identifies the key variables through the MICMAC method.

The MICMAC analysis was carried out on "Micmac", which is a software by the French institute of IT Innovation for the Enterprise under the supervision of its conceptual creator LIPSOR, the Laboratory of Innovation of Strategic Prospective and Organization, and calculates the impact of previously identified relationships and thus can prioritize the variables from the structural analysis matrix. Duperrin and Godet [46] provided the following definition: "the MICMAC is a matrix multiplication program applied to the structural matrix which makes it possible to study the diffusion of impacts through the paths and feedback loops, and consequently prioritizes the variables".

However, the initial structural analysis matrix has been established only from direct relationships between the variables. In fact, a variable can also exert its influence on other variables indirectly, either through the intermediary of another variable ("path" of order 2), or the intermediary of several others exerting their influence in cascade, through increasingly long "paths", which can also loop on themselves.

The MICMAC has been used in different fields to prioritize and determine the key variables in different areas. In SCRM, MICMAC analysis should quantify and classify the risk variables based on their influence and dependence on other risk variables and highlight counter-intuitive risk variables. The aim behind this is to recognize the influential risk variables and the dependent risk variables, and to find out the key risk variables and their relationships. These prioritized risk variables provide a useful tool for supply network managers to focus on those key variables for effective risk mitigation strategies.

### 2.2. The Proposed Approach

The primary aim of this research paper was to study the relationship between the SDP, the risk management and performing SCs. Indeed, the SDPs were classified according to their priority for better risk management and effective SC performance. As described in Figure 1, the proposed approach contains two phases. First, 14 SDPs were identified and selected from the literature. Second, the MICMAC method as a structural analysis method was applied to identify and assess sustainable practices which were more responsible for reducing risk in the SC. The input data for each phase, which was based on Delphi technique, was a group of processes used to survey and collect the opinions of experts on a particular subject. Based on the literature review and discussion with experts, researchers and academicians in Delphi analysis, the following two steps of the proposed approach were applied:

In the first step, 26 sustainable management practices were identified from the literature review. After that, only 14 practices were analyzed in this study, as founded by Jellali and Benaissa [15]. The second step was about the identification of the relationship between the practices selected through a structural analysis matrix. The filling of this matrix was done by the experts' agreement about the determination of the influence degree. Based on this matrix, the prioritization of the SSCRM practices was performed using the MICMAC method.

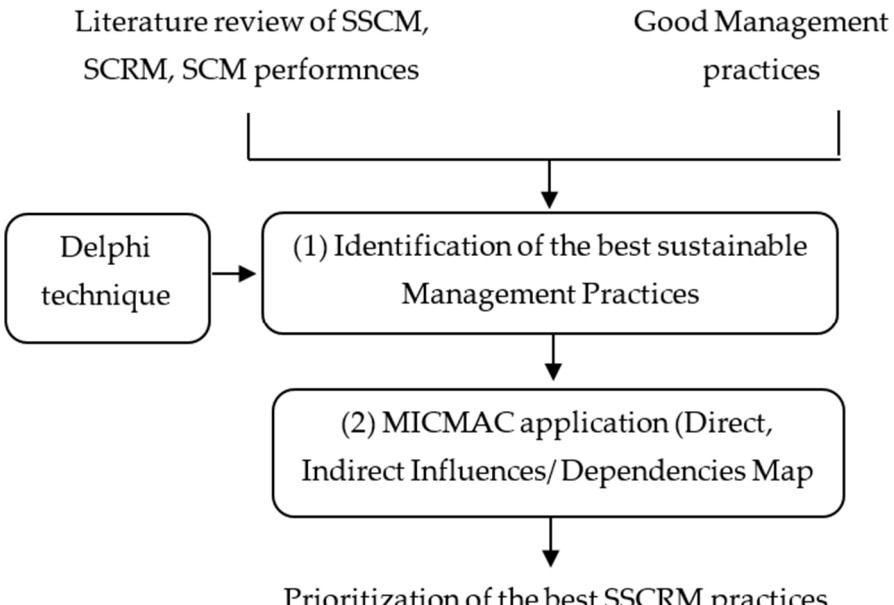

**Figure 1.** The flowchart of the proposed approach.

The MIMAC method was applied using a computer software called "MICMAC", which was developed by the French Computer Innovation Institute "3 IE" under the supervision of its conceptual creators LIPSOR laboratory. The structural analysis matrix, also called the direct influences/dependencies matrix, was the input of the MICMAC software that generated the key sustainable management practices by evaluating the direct and indirect influences/dependencies of the practices among each other.

The motivation of this work was to determine the key best sustainable management practices by studying their inter-relationships, influences and dependencies. This was done using the MICMAC method. The choice of the 14 selected practices was based on their sustainability aspect. The simple application of the MICMAC method on the 26 practices did not consider their three (economic, social and environmental) dimensions of sustainable development on which the authors carried out their preliminary evaluation.

### 3. Results

*3.1. Identification of the Best Sustainable Management Practices*

In fact, the Table 1 presents the 26 SSCM practices identified in the literature [15,47]. In this context, Bauman [47] found an interesting database of good practices commonly implemented in the SCM. All the selected practices are recommended by at least one SCM standard. The well-known three models related to the SCM are selected [48]. The first one is the famous SCOR model which insists on the functions and the performance indicators for the supply chain. The second reference is the ASLOG framework and finally, the third reference is EVALOG.

In fact, Table 1 does not split the practices into the environmental, economic and social aspects because all or some areas may be included in each practice. Given that the SDP refers to the threefold of the environmental performance, social responsibility and economic profitability and the performance indicators chosen in this research focus on the three dimensions of sustainability.

The final 14 selected sustainable management practices are: P1, P2, P3, P4, P5, P6, P7, P8, P9, P10, P11, P12, P16 and P19. The other following P13, P14, P15, P17, P18, P20, P21, P22, P23, P24, P25 and P26 practices will not intervene in the analysis since they have no impact on the selected sustainable performance indicators and therefore, they will not be presented in this work.

**Table 1.** Presentation on the selected SSCM practices.

| Process | Practices | Reference |
|---|---|---|
| DESIGN | P1. Delayed differentiation | SCOR |
| | P2. Sharing with upstream and/or downstream partners knowledge or ideas around the design of new products. | SCOR, EVALOG, ASLOG |
| | P3. Simplification of the dismantling of products/Anticipation of the end of the product life. | SCOR, EVALOG, ASLOG |
| BUY | P4. Evaluation of suppliers/subcontractors. | SCOR |
| | P5. Selection of suppliers/subcontractors according to geographic criteria in order to promote local businesses. | SCOR, EVALOG, ASLOG |
| | P6. Work with local buyers. | SCOR |
| SUPPLY | P7. Replenishment via a kanban system. | SCOR, EVALOG, ASLOG |
| | P8. Pooling of supplies for several suppliers or service providers. | SCOR |
| | P9. Establishment of shared management of supplies (CPFR). | SCOR |
| SALE | P10. Establishment of a system allowing to show the conditions under which a product promised to a customer can be delivered. | SCOR |
| | P11. Building relationships with customers to better understand them in order to adapt and personalize products/services. | SCOR, EVALOG, ASLOG |
| PRODUCE | P12. Lean Manufacturing approach. | SCOR |
| | P13. Combination of the three SCOR methodologies/Six Sigma/Lean Manufacturing. | SCOR |
| | P14. Cadence-buffer-link technique. | SCOR |
| DISTRIBUTE | P15. Consolidation of transport needs by client, source, route, intermediaries. | SCOR, EVALOG, ASLOG |
| | P16. Establishment of contracts with Carriers. | SCOR |
| | P17. Establishment of cross-docking. | SCOR |
| | P18. Implementation of a planning by wave of Sample. | SCOR |
| RETURN | P19. Management of return logistics. | SCOR, EVALOG, ASLOG |
| | P20. Sorting of waste. | SCOR, EVALOG, ASLOG |
| DIRECTION | P21. Implementation of planning tools in Network. | SCOR |
| | P22. Supply chain risk management. | SCOR, EVALOG, ASLOG |
| | P23. Establishment of shared management of procurement. | SCOR |
| | P24. Stock management method "Statistical Test Count". | SCOR |
| SUPPORT | P25. Management of electronic reminders for maintenance deadlines | SCOR, EVALOG, ASLOG |
| | P26. Installation of electronic transmission system. | SCOR, EVALOG, ASLOG |

*3.2. MICMAC Application*

After having sought exhaustiveness in the list of practices to be considered, it is now reducing the complexity of the problem and detecting which key variables should be investigated as a priority. The practices that do not seem to play on the studied system could be neglected. Then, the MICMAC aim is to identify the most influential practices and more dependent (the key practices), by constructing a typology of practices in direct and indirect classification.

In this context, Godet [42] specifies that the "MICMAC (Matrix of Crossed Impacts—Multiplication Applied to a Ranking) is a matrix multiplication program applied to the structural matrix. Then, the MICMAC software distinguishes between the direct influence matrix (DIM) and the indirect influences matrix (IIM).

The input matrix of the MICMAC method is the DIM. As mentioned in Table 2, this matrix contains values between 0 and 3 which shows the direct influence relationship between the variables defining the system. Therefore, the 0 value shows no influence, value of 1 shows a weak influence, value of 2 is used for a medium influence while value 3 shows a powerful influence. We should note that the DIM matrix is not a symmetric matrix.

3.2.1. Direct Influence/Dependencies Practices Map

One output of the MICMAC software is the Direct/Dependencies map, as mentioned in Figure 2. It is determined from the DIM, whose x-axis represents the dependence, and the y-axis represents the influence.

**Table 2.** Direct Influence Matrix.

|  | P1 | P2 | P3 | P4 | P5 | P6 | P7 | P8 | P9 | P10 | P11 | P12 | P16 | P19 |
|---|---|---|---|---|---|---|---|---|---|---|---|---|---|---|
| **P1** | 0 | 1 | 0 | 0 | 2 | 2 | 1 | 3 | 1 | 1 | 3 | 2 | 3 | 3 |
| **P2** | 2 | 0 | 1 | 2 | 3 | 3 | 2 | 3 | 2 | 2 | 3 | 3 | 3 | 3 |
| **P3** | 1 | 0 | 0 | 1 | 3 | 3 | 2 | 3 | 2 | 2 | 3 | 2 | 3 | 3 |
| **P4** | 1 | 0 | 0 | 0 | 3 | 3 | 2 | 3 | 2 | 1 | 3 | 2 | 3 | 3 |
| **P5** | 0 | 0 | 0 | 0 | 0 | 1 | 0 | 2 | 0 | 0 | 0 | 0 | 2 | 1 |
| **P6** | 0 | 0 | 0 | 0 | 0 | 0 | 0 | 1 | 0 | 0 | 0 | 0 | 1 | 2 |
| **P7** | 0 | 0 | 0 | 0 | 0 | 1 | 0 | 2 | 0 | 0 | 1 | 1 | 0 | 2 |
| **P8** | 0 | 0 | 0 | 0 | 0 | 0 | 0 | 0 | 0 | 0 | 0 | 0 | 2 | 0 |
| **P9** | 1 | 0 | 0 | 0 | 0 | 0 | 0 | 0 | 0 | 0 | 0 | 0 | 0 | 0 |
| **P10** | 0 | 0 | 0 | 2 | 2 | 2 | 1 | 2 | 0 | 0 | 1 | 1 | 3 | 3 |
| **P11** | 1 | 0 | 0 | 0 | 2 | 2 | 1 | 3 | 1 | 1 | 0 | 3 | 3 | 3 |
| **P12** | 0 | 0 | 0 | 0 | 2 | 2 | 3 | 3 | 1 | 1 | 1 | 0 | 3 | 2 |
| **P16** | 0 | 0 | 0 | 0 | 0 | 0 | 0 | 0 | 0 | 0 | 0 | 1 | 0 | 0 |
| **P19** | 0 | 0 | 0 | 0 | 0 | 0 | 1 | 0 | 0 | 0 | 0 | 0 | 2 | 0 |

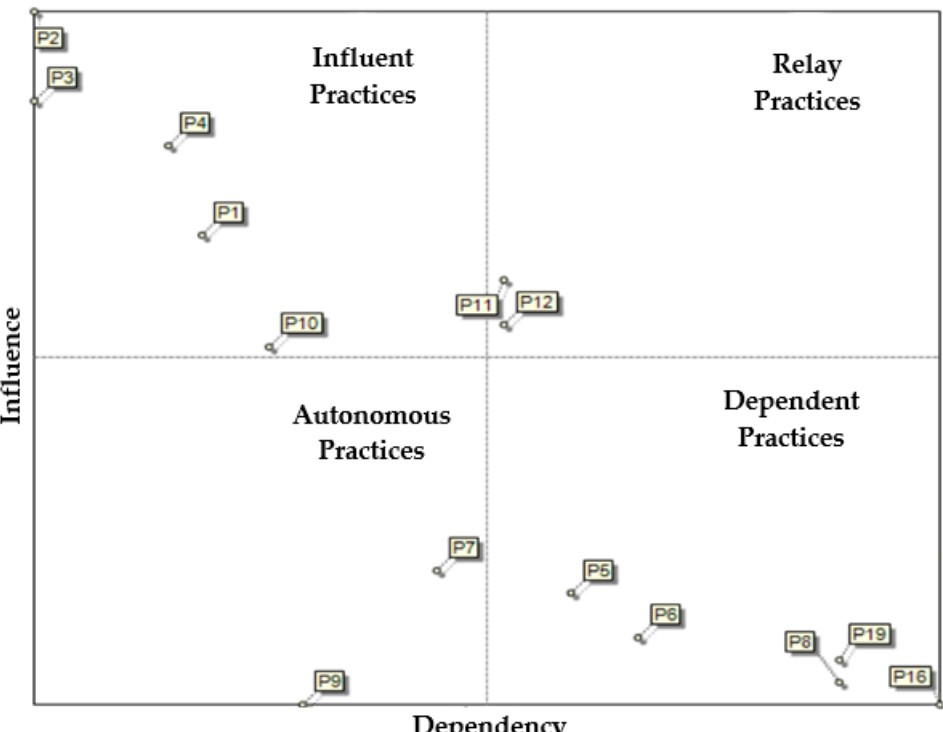

**Figure 2.** Direct Influence/Dependencies practices Map.

The map stated in Figure 2 contains four regions (quadrants) according to practices of direct influence and direct dependency. The first region contains autonomous practices such as P7 and P9. Autonomous variables, which they drop from the analysis, have no influence or dependency. The second region (Influent practices) contains the most influential practices such as P1, P2, P3, P4 and P10. In this third region, the practices can be input practices. The third region (dependent practices) concerns practices with a high direct dependency. This region contains P5, P6, P8, P16 and P19. Practices of this quadrant are output practices because they have a low influence but are strongly influenced by other practices. Finally, the fourth region contains relay practices which have a high dependency

and a high influence. In this study, there are only two relay practices (key practices) which have a high dependency and a high influence. Practices belonging to this quadrant play the role of the input practices and output practices at the same time. These two relay practices are (P11) "Building relationships with customers to better understand them, adapt and personalize products/services" and (P12) "lean manufacturing approach".

The input practice (P2): sharing with upstream and/or downstream partners of knowledge or ideas around the design of a new product, and the practice (P3): simplification of product dismantling/anticipation of end of life of Products, are very influential but not very dependent practices. The applications of practices P2 and P16 influence the application of all the other practices.

Finally, the results of the dependent practices (P16): establishment of contracts with carriers, (P8): pooling of supplies from several suppliers or service providers and (P19): management of return logistics" are not very influential and very dependent. They are output practices. These three practices with autonomous practices (P7 and P9) should not have priority application for risk management.

3.2.2. Indirect Influences/Dependencies Practices Map

The IID matrix corresponds to the DIM raised in power, by successive iterations. From this matrix, a new classification of the variables highlights the most important practices. In fact, hidden practices are detected, thanks to a matrix multiplication program applied to an indirect classification. This program makes it possible to study the diffusion of the impacts by the paths and the feedback loops, and consequently to rank the practices: in an order of influence while considering the number of paths and loops of length 1, 2, ... n from each variable; in an order of dependence by considering the number of paths and the loops of length 1, 2, ... n arriving at each practice. The classification becomes stable from a multiplication to the order of 4 or 5 [42]. Then, the got indirect influences/dependencies map is showed in Figure 3.

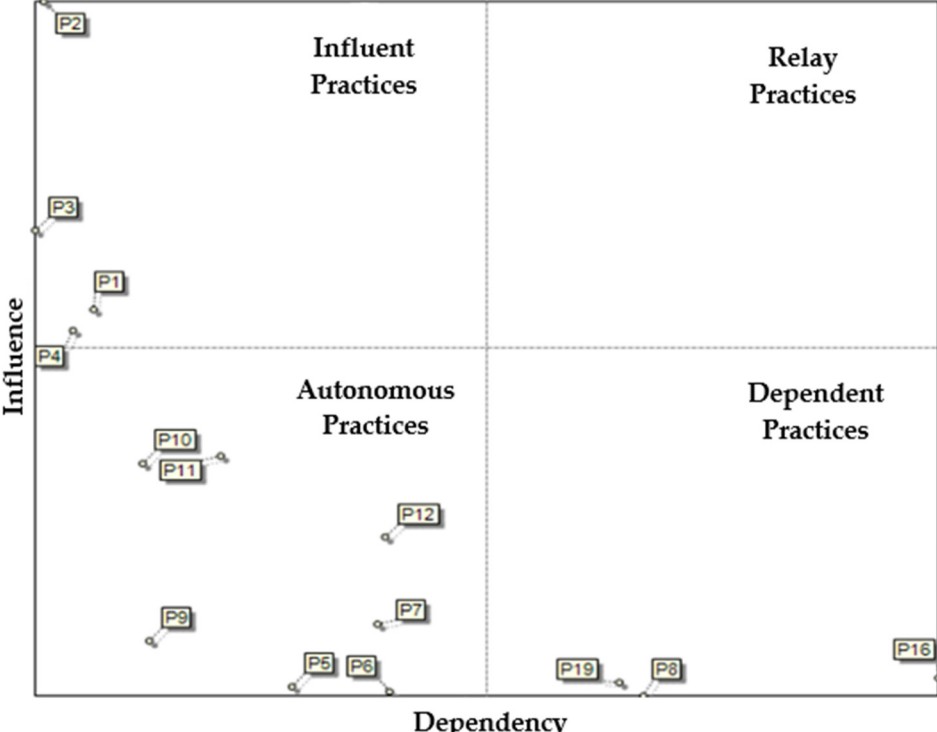

**Figure 3.** Indirect Influences/Dependencies Map.

The map stated in Figure 3 also contains four quadrants, as in Figure 2 according to the practices indirect influence and indirect dependency. Fortunately, in this case there

are no relay practices so there is stability of the set of the studied practices. The input practices (P1): delayed differentiation, (P2) sharing with upstream and/or downstream partners of knowledge or ideas around the design of a new product, (P3) simplification of the dismantling of products/anticipation of the end of product life and (P4) evaluation of suppliers/subcontractors are very influential practices and not very dependent on the other practices.

The results of dependent practices are (P16): establishment of contracts with carriers, (P8): pooling of supplies from several suppliers or service providers, (P19): management of return logistics" are not very influential but very dependent.

3.2.3. Classification of Practices According to Their Influences/Dependencies

A basic result from the MICMAC method application is the acquired numerical weights (direct influences/dependencies and indirect influences/dependencies) of all the studied practices and classifies them as motioned in Table 3, in a descending order. The SSCRM practices are classified through four criteria include the following elements: (1) their direct influences classification, (2) their indirect influences classification, (3) their direct dependencies classification and (4) indirect dependencies classification.

**Table 3.** Numerical weights (direct influences/dependencies and indirect influences/dependencies) of the practices.

| Rang | Practice | Direct Influence | Practice | Direct Dependencies | Practice | Indirect Influence | Practice | Indirect Dependencies |
|---|---|---|---|---|---|---|---|---|
| 1 | P2 | 1711 | P16 | 1497 | P2 | 2474 | P16 | 2300 |
| 2 | P3 | 1497 | P8 | 1336 | P3 | 1669 | P8 | 1550 |
| 3 | P4 | 1390 | P19 | 1336 | P1 | 1391 | P19 | 1488 |
| 4 | P1 | 1176 | P6 | 1016 | P4 | 1317 | P6 | 904 |
| 5 | P11 | 1069 | P5 | 909 | P11 | 876 | P12 | 893 |
| 6 | P12 | 962 | P11 | 802 | P10 | 849 | P7 | 874 |
| 7 | P10 | 909 | P12 | 802 | P12 | 591 | P5 | 656 |
| 8 | P7 | 374 | P7 | 695 | P7 | 284 | P11 | 475 |
| 9 | P5 | 320 | P9 | 481 | P9 | 225 | P9 | 294 |
| 10 | P6 | 213 | P10 | 427 | P16 | 95 | P10 | 277 |
| 11 | P19 | 160 | P1 | 320 | P19 | 79 | P1 | 152 |
| 12 | P8 | 106 | P4 | 267 | P5 | 64 | P4 | 100 |
| 13 | P9 | 53 | P2 | 53 | P6 | 47 | P2 | 25 |
| 14 | P16 | 53 | P3 | 53 | P8 | 32 | P3 | 4 |

Table 3 also shows the ranking change on the one hand, between direct and indirect influence and between direct and indirect dependency. Practices P1, P2, P3, P4, P11, P12 and P10 have the highest influence practices for both direct and indirect manners. These practices are considered the inputs practices and their applications, which can improve the application of all the other studied practices.

Practices P5, P6, P8, P11, P12, P16 and P19 have the most dependency practices for both direct and indirect manners. Caution will be made for relay practices P11 and P12 which are input and output practices for direct influence.

## 4. Discussion

As a summary of the got results, Table 4 shows the classification of the most important practices. In fact, four sets are got according to direct, indirect influence and dependency. The intersection of theses sets, which provides the key practices, is also mentioned in Table 4.

The most influential SD practices are P1, P2, P3 and P4. They should be the first practices that must attract the attention of managers in the supply chain. In addition, P9, P7 and P12 have raised an indirect ranking. So that managers should consider their important indirect influencing aspect. This means that following these practices will indirectly have a positive effect on the SC performance and therefore, should be responsible and not neglected.

**Table 4.** Numerical weights (direct influences/dependencies and indirect influences/dependencies) of the practices.

|  | Dependencies | Influences |
|---|---|---|
| Direct | P5, P6, P8, P16, P19, P11, P12 | P1, P2, P3, P4, P10, P11, P12 |
| Indirect | P8, P16, P19 | P1, P2, P3, P4 |
| Intersection | P8, P16, P19 | P1, P2, P3, P4 |

The six key findings SSCRM practices from direct and indirect classification include the following elements: (1) delayed differentiation, (2) information sharing with upstream and/or downstream partners, (3) simplification of product dismantling/anticipation of product end of life, (4) supplier/subcontractor performance assessment, (5) establishing shared supply management and (6) establishment of contracts with transporters.

Even if its results agree with existing findings of Jellali and Benaissa [15], which identify the key practices, this work results outline the importance of the following practices: P1, P10 and P16 according to its important indirect influences. The role of P12 and P7 are exerted by their indirect dependency.

Therefore, compared to previous works, the proposed approach has many strong points such as (1) stimulating reflection within the group and to make people think about counter-intuitive aspects of the behavior of a system, (2) asking the right questions to get good answers and reduce inconsistencies in reasoning, (3) helping the decision-maker, rather than replacing him. It does not attempt to accurately describe how the system works, but its purpose is to identify the main organizational elements of the system, and (4) highlighting the influential and dependent practices.

Despite its effeteness, this method suffers from limitations because of the subjective nature of the list of variables, and the subjective character of the filling of the matrix (notation of relations). A matrix is never a reality, but a way of looking at it, a photograph. Like any photograph, structural analysis shows things that reflect part of the reality.

## 5. Conclusions

The term sustainable development and risk management have gradually entered the supply chain lexicon. Therefore, this research work tackles the problem about, "how to build a risk reduction strategy that takes sustainable development into account?"

The relationship between sustainable development practices and risk management has been sought to identify the practices that reduce more risks in the supply chain. For this reason, the MICMAC method applied to understand the direct and indirect relationships between the various good management practices in an extended supply chain and determine which ones are of higher priority and which further reduce the risks in the supply chain. The results got by the MICMAC method have shown that only four practices are important, which should attract the attention of managers when managing risks in the supply chain. These practices are "Delayed differentiation (P1)", "Sharing with upstream and/or downstream partners of knowledge or ideas around the design of a new product (P2)", "Simplification of product dismantling/Anticipation of Product end of life (P3)" and "Supplier/subcontractor performance assessment (P4) are the most critical variables for developing risk mitigation strategies. The analysis also highlights a significant reflect to practices that have important indirect ranking which are (P9) establishing shared supply management and (P16) establishment of contracts with transporters.

This research has three contributions. First, it prioritizes the best SSCM practices that can be taken as avoiding supply chain risk mitigation strategies for an effective whole supply chain performance and gives more weight to the counter-intuitive results. Second, this research stimulates reflection within the group. Finally, it helps the leader to elaborate supply chain risk strategies in a prospective view by considering the best key sustainable risk practices, instead of replacing it.

To better improve this research work, it is necessary to apply the results to a real case study and see how these emerging sustainable practices really contribute to a better

risk management in the treated logistics chain. Thus, future directions are related to the improvement of these results using a multi-criteria decision-making method to compare the best sustainable risk management practices among different supply chain actors' policies and assess the supply chain resilience from different supply chain stakeholders' perspectives.

**Author Contributions:** Conceptualization, M.E. and W.H.; methodology, M.E., W.H. and A.M.A.; validation, W.H. and A.M.A.; formal analysis, M.E. and W.H.; investigation, M.E.; resources, M.E.; data curation, M.E. and W.H.; writing—original draft preparation, M.E.; writing—review and editing, W.H. and A.M.A.; visualization, A.M.A.; supervision, W.H.; project administration, W.H. and A.M.A.; funding acquisition, A.M.A. All authors have read and agreed to the published version of the manuscript.

**Funding:** This research was supported and funded by Taif University Researchers Supporting Project number (TURSP-2020/229), Taif University, Taif, Saudi Arabia. The authors are grateful for this financial support.

**Institutional Review Board Statement:** Not applicable.

**Informed Consent Statement:** Not applicable.

**Data Availability Statement:** Data is contained within the article.

**Acknowledgments:** This research was supported by the Taif University Researchers who supported Project number (TURSP-2020/229), Taif University, Taif, Saudi Arabia. First, the authors are grateful for this financial support. Second, the authors would like to express their thanks to all the experts for their help and their implication during the Delphi technique application. Finally, the authors would like to thank the editor and the anonymous reviewers, whose insightful comments and constructive suggestions helped us to significantly improve the quality of this paper.

**Conflicts of Interest:** The authors declare no conflict of interest.

## Abbreviations

Notations: The following acronyms are used in this manuscript:

| | |
|---|---|
| ASLOG | Association Française pour la Logistique |
| DEMATEL | Decision making trial and evaluation laboratory |
| DIM | Direct Influences Matrix |
| EVALOG | Evaluation Logistics-Materials Management Operations |
| IIM | Indirect Influence Matrix (IIM). |
| ISM | Interpretive Structural Modelling |
| MICMAC | Matrice d'Impacts Croisés Multiplication Appliquée à un Classement (French) |
| MID | Direct Influences Matrix |
| IIM | Indirect Influences Matrix |
| SC | Supply Chain |
| SCM | Supply Chain Management |
| SCOR | Supply Chain Operations Reference |
| SCRM | Supply Chain Risk Management |
| SDP | Sustainable Development Practices |
| SSC | Sustainable Supply Chain |
| SSCM | Sustainable Supply Chain Management |
| SSCRM | Sustainable Supply Chain Risk Management |
| UNCED | United Nations Conference on Environment and Development |

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
