# Peer review of "Prioritization of the Best Sustainable Supply Chain Risk Management Practices Using a Structural Analysis Based-Approach"

_sustainability, doi:10.3390/su13094608_

Round 1

Reviewer 1 Report

The paper is well prepared. I found some problems in the text, but they do not they do not lower the substantive value of the text:

  1. In many places (p. 2, l. 49; p.5, l. 221; p 6, line 275, 281, 284; p. 12, l. 494; p. 13. l. 533, 547; p. 14. l. 586) you use the text: "the following. (1)". Is it suggestion but I thunk you should use ":" instead of full stop, because after it you list elements
  2. page 10, table 1: First line, problem with reading (quality of letters).   I know that they were created in programm, but maybe in can be improved. In addition, the text in first line should be rotated by 180 degrees.
  3. Figures 2 (page 10)  & 3( page 11) - you have text in French. It is better to translate it into English. I know that they were created in programme, so maybe you should add explanation.
  4. Page 13, line 547 and page: "relationships. this work" - start "this" with capital letter
  5. Page 15, line 589: "group, (3) help". In other number (1, 2) you start with capital letter, so here also should "Help"
  6. Problems with refefences:
  • Literature 26, 28, 30,34, 42 (wrong method of writing the notes - different than others),
  • literature: 14, 15, 16, 75, 47 - year of publication should be prepared in bold

Author Response

The authors would like to thank the editor and the anonymous reviewers, whose insightful comments and constructive suggestions helped us to significantly improve the quality of this paper. Every change in the text is colored in red

Reviewer 2 Report

Problem 1: There are many minor spelling mistakes and grammatical mistakes in the paper. For example,

Line 42: Use investigations instead of investigates (There are many such typos and language errors. Please review thoroughly before publication)

Line 68, Line77/78, LIne 143, Lin 213 all have errors.

Problem 2: Page 8 talks about identification of best sustainable  management practices. Please separate the environmental, economical and social aspects in the table. OR if all or some are included, please state so.

Author Response

(The authors gave the same response as above.)

Reviewer 3 Report

Work written correctly, correctly selected drawings, correct interpretation of the results. With regard to the literature, in the opinion of the reviewer, the literature review is too extensive, I suggest limiting the items to 30.

Author Response

(The authors gave the same response as above.)

Round 2

Reviewer 2 Report

The spelling and grammatical mistakes still exist in the document. Need to be reviewed and edited with the Journal resources before publication.

Author Response

The authors would like to thank the editor and the anonymous reviewers, whose insightful comments and constructive suggestions helped us to significantly improve the quality of this paper.  

The manuscript is corrected as much as possible 
